# A Multi-Omics Approach Revealed Common Dysregulated Pathways in Type One and Type Two Endometrial Cancers

**DOI:** 10.3390/ijms242216057

**Published:** 2023-11-07

**Authors:** Valeria Capaci, Lorenzo Monasta, Michelangelo Aloisio, Eduardo Sommella, Emanuela Salviati, Pietro Campiglia, Manuela Giovanna Basilicata, Feras Kharrat, Danilo Licastro, Giovanni Di Lorenzo, Federico Romano, Giuseppe Ricci, Blendi Ura

**Affiliations:** 1Institute for Maternal and Child Health, IRCCS Institute for Maternal and Child Health-IRCCS Burlo Garofolo, 34137 Trieste, Italy; valeria.capaci@burlo.trieste.it (V.C.); michelangelo.aloisio@burlo.trieste.it (M.A.); feras.kharrat@burlo.trieste.it (F.K.); giovanni.dilorenzo@burlo.trieste.it (G.D.L.); federico.romano@burlo.trieste.it (F.R.); giuseppe.ricci@burlo.trieste.it (G.R.); blendi.ura@burlo.trieste.it (B.U.); 2Department of Pharmacy, University of Salerno, 84084 Salerno, Italy; esommella@unisa.it (E.S.); esalviati@unisa.it (E.S.); pcampiglia@unisa.it (P.C.); mbasilicata@unisa.it (M.G.B.); 3AREA Science Park, Basovizza, 34149 Trieste, Italy; licastrod@gmail.com; 4Department of Medicine, Surgery and Health Sciences, University of Trieste, 34149 Trieste, Italy

**Keywords:** multi-omics, mass spectrometry, endometrial cancer

## Abstract

Endometrial cancer (EC) is the most frequent gynecologic cancer in postmenopausal women. Pathogenetic mechanisms that are related to the onset and progression of the disease are largely still unknown. A multi-omics strategy can help identify altered pathways that could be targeted for improving therapeutical approaches. In this study we used a multi-omics approach on four EC cell lines for the identification of common dysregulated pathways in type 1 and 2 ECs. We analyzed proteomics and metabolomics of AN3CA, HEC1A, KLE and ISHIKAWA cell lines by mass spectrometry. The bioinformatic analysis identified 22 common pathways that are in common with both types of EC. In addition, we identified five proteins and 13 metabolites common to both types of EC. Western blotting analysis on 10 patients with type 1 and type 2 EC and 10 endometria samples confirmed the altered abundance of NPEPPS. Our multi-omics analysis identified dysregulated proteins and metabolites involved in EC tumor growth. Further studies are needed to understand the role of these molecules in EC. Our data can shed light on common pathways to better understand the mechanisms involved in the development and growth of EC, especially for the development of new therapies.

## 1. Introduction

Endometrial cancer (EC) is the most common gynecologic malignancy. It arises from the endometrium, the inner part of the uterus. It is the third most common cancer in women in North America and Western Europe, and its incidence is still increasing [1,2]. EC risk factors are related to increasing age; obesity; hypertension; metabolic diseases, including diabetes mellitus; and, most importantly, the exposure of the endometrium to high estrogen levels [3,4,5]. Additional risk factors are related to genetic predisposition and ancestral origin, with black women being more prone to this type of cancer [6]. Two types of EC can be distinguished: type 1, estrogen-dependent, comprising 80% of cases with good prognosis, and type 2, estrogen-independent, with a worst prognosis [7,8]. At present, no diagnostic test is available for ECs, and the mainstay of treatment is total hysterectomy [9]. In addition, the pathogenic mechanisms responsible for EC are still to be identified. A wider understanding of the molecular processes involved in ECs could help to improve diagnosis, patient stratification and treatment. In the last 10 years, ”omics strategies have been increasingly used to understand biological processes underlying diseases and to discover new biomarkers associated with diagnosis, progression and response to treatment” [10].

Multi-omics data from cancer cells, are essential to better understand the pathophysiology of tumors and the response to drugs, to develop more targeted therapies. The development of omics and new bioinformatic tools has given extraordinary opportunities to further classify cancers into subtypes, identify new biomarkers and improve survival rates [11].

Therefore, to improve the knowledge of EC, we took advantage of different ”omic strategies” to explore the EC landscape. In particular, the focus was placed on transcriptomic, proteomic and metabolomic lipidomics analyses representing all the transcripts, proteins and metabolites, respectively, expressed in a cell, a tissue or an organism [12,13,14,15]. Proteomic analyses study the abundance, the variety of proteoforms or the post-translational modifications of all proteins of a specimen. Various proteomic approaches have been used in cancer research for the identification of specific biomarkers or the molecular players involved in tumorigenesis [16]. Additionally, the relevance of cell metabolism and its derangement in cancer development and progression is also emerging [17]. Metabolomic analyses deal with the quantitative and qualitative assessment of metabolites, representing the intermediates or the end products of metabolism [18]. Compared to mRNAs and proteins, metabolites reflect considerably small modifications in cell behavior and might help unveil subtle processes and pathways involved in cancer biology. In addition, thanks to databases and prediction tools, such as Reactome, KEGG, IPA, and MetaboAnalyst, it is possible to predict the pathways active in different diseases and conditions based on the expression of mRNAs, proteins and metabolites [19,20], obtaining more information about the molecular basis of tumorigenesis and helping in the development of selective anticancer therapies [18].

In this work, by exploiting different cellular models, we applied a multi-omics approach to gain new insight into EC biology, aiming to identify new pathways shared by EC type 1 and 2 that can help us decipher the pathological processes in EC.

## 2. Results

### 2.1. Metabolomics

To conduct multi-omics analyses, a triple biological replicate was conducted on cell lines. Metabolomics analyses of four EC cell lines (ANCA (type 1 metastatic), ISHIKAWA (type 1 non-metastatic), KLE (metastatic type 2), HEC1A (non-metastatic type 2)) were carried out by HILIC-HRMS. 132 metabolites were annotated at MSI level 2. We made four comparisons between type 1 and type 2 EC cell lines to identify dysregulated metabolites (fold change ≥ 1.5 or ≤0.69 (*t*-test *p*-value < 0.05) (Table 1) (metabolomics data available in Appendix A).

For the identification of common dysregulated metabolites present in all four cell lines, we performed a Venn Diagram data analysis. (Figure 1) The Venn Diagram was generated using data in Table 1.

This analysis revealed that 13 metabolites (Xanthine, N-Acetylaspartic acid, Methyl-2-hydroxyisobutyric acid, FA 20:4, Butyryl carnitine, Acetylcarnitine, Fumaric acid 4-Aminobutyric acid, Cystathionine, Glucosamine-6-phosphate, S-Adenosylhomocysteine and Threonine Methionine sulfoxide) were dysregulated in all four comparisons. Being that in the first analysis with the four comparisons the number of metabolites was low, we conducted another analysis including three comparisons of four EC cell lines (i.e., ANCA vs. ISHIKAWA, KLE vs. ANCA, and KLE vs. HEC) which revealed 32 dysregulated metabolites (Indoleacetic acid, Pyroglutamic acid, Guanosine, Nicotinamide, Uric acid, 5′-Methylthioadenosine, Vaccenic acid, Glyceric acid, Proline, Methionine, Norleucine, Inosine, Glucose phosphate, Carnosine, D-Ribose 5-phosphate, Alanine, N-Acetylaspartylglutamic acid, Uridine, Hydroxyphenyllactic acid, Hypoxanthine, N-Acetyl-methionine, Adenosine diphosphate, Carnitine, Cytidine, Cytosine, N-Acetylneuraminic acid, Propionylcarnitine, Fructose, Phenylalanine, N6-Acetyllysine and N-Acetylysine). Table 2 summarizes the metabolite abundance trend in both analyses.

The bioinformatics tool MetaboAnalyst 5 was used for the pathway enrichment (Figure 2) and chemical main class identification (Figure 3) of these 45 dysregulated metabolites that belong to the metabolic pathway: amino acids metabolism, fatty acid-branched and very-long-chain, and amino sugar metabolism.

Regarding the main chemical classes, these are categorized into the following: amino acids and peptides, purines, monosaccharides and glycosyl compounds.

### 2.2. Proteomics

For the whole proteomics investigation, the cellular digestion labeled with the TMT 2plex chemical label was subjected to nanoLC–MS/MS-based proteomic analysis. Through the bioinformatic analysis, using Proteome Discoverer 2.5 combining Precursor Detector and Sequest HT nodes, we identified 15,028 proteins with FDR < 1%. Out of these, 2669 with q-value < 0.05 and FDR < 1% were used for further analysis. We made the same cell comparison as in metabolomic analysis to identify dysregulated proteins (fold change ≥ 1.2 or ≤0.85 *t*-test *p*-value < 0.05) (Table 3) (Proteomics data available in Appendix A).

To identify proteins in common between the various pairings, we have built a Venn Diagram (Figure 4) using data from Table 3. The Venn Diagram data analysis of all four comparations showed five common proteins (CFL1, STRAP, ART3, ACAT2 and NPEPPS) dysregulated in both types of EC cell lines. Another analysis with three comparisons, including A vs. I; K vs. A; and K vs. H, showed 19 common dysregulated proteins (EEF2, RTRAF, PSMA5, IPO5, CNBP, AP2B1, CLTC, RUVBL2, ARHGDIA, SNRPB, S100A11, ANP32E, PA2G4, PTMA, RPN2, PCBP2, ANXA2, U2AF2 and CAPS). Table 4 summarizes the protein abundance trend in both analyses.

We used the gProfiler tool for protein classification (Figure 5) and were able to classified the 24 proteins into groups according to their molecular function and cellular component. Regarding molecular function, proteins were categorized into the following: U2 snRNP binding, snRNP binding, ribonucleoprotein complex binding and S100 protein binding. In terms of cellular components, proteins were categorized into the following: extracellular exosome, extracellular vesicle, extracellular organelle, extracellular membrane-bounded organelle, vesicle, vesicle lumen, extracellular space, cytosol and secretory granule lumen.

IPA tool showed (Figure 6) that these proteins are set on in top networks corresponding to the following: (A) the apoptosis of tumor cell lines (ANXA2, ARHGDIA, CLTC, PCBP2, PSMA5, PA2G4, PTMA, S100A11 and SNRPB); (B) the cell death of tumor cells (ANXA2, PSMA5, SNRPB and U2AF2); (C) the migration of tumor cell lines (ANXA2, CLTC, PA2G4, RUVBL2, S100A11 and U2AF2); and (D) the splicing of mRNA (SNRPB, STRAP and U2AF2).

The next bioinformatics analysis was for the identification of pathways disrupted in common between EC cell types 1 and 2. For this purpose, the MetaboAnalyst tool was used to link proteomics data with metabolomics data. We identified 22 pathways common to EC type 1 and type 2 (*p*-value < 0.05) (Table 5).

### 2.3. Western Blotting

For the validation of the dysregulated protein NPEPPS identified by LC/MS-MS, we performed Western blotting on 10 EC vs. 10 healthy controls (Figure 7). We tried to validate NPEPPS because of its up-regulation in all the pairings. A statistically significant higher abundance was found (Mann–Whitney sum-rank test *p* < 0.05) for NPEPPS (*p* = 0.02).

## 3. Discussion

In this study, we applied a multi-omics approach for the study of four type 1 and 2 metastatic and non-metastatic EC cell lines for common pathway identification. At last, from the five common proteins identified by bioinformatic tools (CFL1, STRAP, ART3, ACAT2 and NPEPPS), we validated NPEPPS. The protein validation was performed by Western blotting on 10 controls vs. 10 patients, showing that proteins are dysregulated not only in cell lines but also in the tissue.

Metabolomics and proteomics are complementary to genetics and can provide relevant information on the pathophysiology of tumors and biomarkers [21]. Metabolomics revolutionized cancer research, through the identification of metabolites whose abundance is affected by disease and can play a role as clinical markers for the identification of disease development and drug response [22]. Raffone et al., in a systematic review based on six studies including 736 patients, concluded that metabolomics can predict the presence of EC and identify the pathological characteristics of the disease [23]. In this study, we have identified several dysregulated metabolic pathways, including not only the amino acids but also the oxidation of fatty acids, as well as the urea cycle, the synthesis of carnitine and the biosynthesis of phosphatidylcholine.

The activation of fatty acid synthesis by cancer cells is essential for the formation of cell components and gene transcription [24]. Other events like oxidation and fatty acid synthesis are essential in the development and proliferation of cancer [14].

The urea cycle is fundamental to eliminate toxic ammonia from the body, and its dysregulation is related to tumor proliferation [25]. In addition, the accumulation of ammonia is toxic for a healthy cell, while cancer cells can recycle ammonia for the synthesis of amino acids [26].

L-carnitine plays a key role in the biosynthesis of fatty acids. Several studies have shown that the carnitine system acts as a block on the metabolic flexibility of cancer cells [27].

Phosphatidylcholine is the main phospholipid component of the eucaryotic membrane and plays an essential role in the proliferation and autophagy of cancer cells, allowing cancer cell survival with limited energetic sources [28].

gProfile analysis revealed that several proteins identified in this study are contained in exosomes. This indicates the possible involvement of these proteins in tumor cell–cell communication. IPA analysis indicates an association between identified proteins and mRNA splicing. Alternative splicing induces regulating cancer cell proliferation by the formation of different isoforms of many cell surface receptors [29].

CFL1 plays a critical role in cell morphology and the cytoskeletal organization of epithelial cells [30]. Bakert et al. demonstrated that the inactivation of this gene leads to the attenuated migration of endometrial cancer cells [31].

STRAP is a receptor-interacting protein that has an important role in cell proliferation, cell death and cancer development [32]. This oncogene induces hepatocarcinoma growth by enhancing Wnt/β-Catenin signaling and regulating β-catenin signaling [33]. Until now, this mechanism has not been completely ascertained.

ACAT2 is a membrane-bound protein involved in the production of cholesteryl ester by using long-chain fatty acyl-CoA and cholesterol as substrates [34]. The expression of this protein varies by the type of tumor [35]. Our lipidomics data indicate a dysregulation of sterol lipids, in which ACAT2 may be involved.

NPEPPS is an aminopeptidase involved in proteolytic events essential for cell growth [36]. This gene is a modulator of the sensitivity of cisplatin in pancreatic cancer [37].

ART4 is an ADP-ribosyltransferase and can modulate the evasion of growth suppressors, cell death resistance and genome instability in cancer [38].

Our bioinformatic analysis highlighted the dysregulation of 23 metabolic pathways associated with tumor growth. The dysregulation of ABC transporter pathways in cancer produces MDR, inducing chemotherapy failure [39]. We identified several amino acid-dysregulated pathways. Amino acids are the essential substrate for protein synthesis, and their metabolism is up-regulated in several cancers. These small molecules facilitate tumor survival in an environment with low Ph and high oxidative stress [40]. Purines are the basic constituents of the nucleotide, and their biosynthesis is mediated by the purinosome. The deregulation of their metabolism is associated with cancer progression [41].

The aminoacyl-tRNA synthetases play a key role in protein synthesis and take part in physiological and tumorigenesis processes [42]. He et al. have identified four SNPs in ARS genes that are associated with an increased risk of breast cancer in the Chinese population [43]. Several studies have shown that ARSs can be considered prognostic or diagnostic biomarkers, and a target of tumor growth and metastasis inhibition [44,45].

Moreover, we are aware that for the identification of common pathways between the two types of ECs, it is necessary to conduct a study on cancer tissue. Another weakness of our study is the limited number of patients used for validation with Western blotting. Further functional studies are also needed to understand the mechanisms of common proteins in EC, such as CFL1, STRAP, ART3, ACAT2 and NPEPPS.

## 4. Materials and Methods

### 4.1. Cell Culture

AN3CA was obtained from ATCC (Cat. n° HTB-111) and cultured in Eagle’s Minimum Essential Medium supplemented with 10% FBS (Fetal Bovine Serum). HEC1A was obtained from ATCC (Cat. n° HTB-112) and cultured in McCoy’s 5a Medium Modified supplemented with 10% FBS. KLE was obtained from IZSLER Lombardia e Emilia Romagna (Cat. n° BSTCL230) and cultured in DMEM/Ham’s F12 medium in a 1:1 ratio with 10% FBS. ISHIKAWA was obtained from CliniSciences (Cat. n° ABC-TC1320) and cultured with Minimum Essential Medium and supplemented with 2 mM Glutamine + 1% MEM NEAA (Minimum Essential Medium Non-Essential AminoAcids) + 5% FBS. All media were supplemented with penicillin and streptomycin (100 IU/mL each). All the cells described were maintained in a 37 °C, 5% CO_2_ incubator.

### 4.2. Patients

During 2021 and 2022, a total of 20 patients (10 women suffering from EC and 10 non-EC controls) were recruited at the Institute for Maternal and Child Health–IRCCS “Burlo Garofolo” (Trieste, Italy). Our Institute’s Technical and Scientific Committee approved the study, and all procedures complied with the Declaration of Helsinki. All patients signed informed consent forms. The median age of patients was 70 years (IQR 57–76, Min = 57, Max = 76). The median age of controls was 44 years (IQR 41–48, Min = 33, Max = 50). Endometrial tissue samples were derived from ordinary leiomyomas in the selection of the controls. We excluded patients and controls with human immunodeficiency virus (HIV), hepatitis B virus (HBV), hepatitis C virus (HCV), leiomyoma and adenomyosis.

### 4.3. Polar and Non-Polar Metabolite Extraction

Polar metabolites were extracted following the MTBE/MeOH/H_2_O based protocol, with slight modifications. Briefly, cell pellets were thawed on ice and treated with 225 µL of ice-cold MeOH containing a mix of deuterated standards. Samples were vortexed for 30 s, incubated for 10 min at −30 °C and sonicated for 10 min at −4 °C. Subsequently, 750 µL of ice-cold MTBE was added, and the solution was continuously agitated in a thermomixer (Eppendorf, Milan, Italy) for 1 h at 500 rpm and at 4 °C. Then, 188 µL of H_2_O was added, and samples were vortexed for 20 s and centrifuged at 14,680 rpm for 10 min at 4 °C to induce phase separation. The upper layer (lipid-containing fraction) was collected and evaporated using a SpeedVac (Savant, Thermo Scientific, Milan, Italy).

For further protein precipitation, ice-cold MeOH was added to the remaining lower phase in a final ratio of 4:1 *v*/*v* MeOH/H_2_O, and the samples were incubated for 1 h at −30 °C, followed by centrifugation for 12 min at 14,680 rpm at −4 °C. The resulting supernatant (polar metabolite-containing fraction) was collected, evaporated using a SpeedVac (Savant, Thermo Scientific, Milan, Italy) and solubilized in 100 µL of ACN/H_2_O (70:30) for metabolomics analysis by HILIC-UHPLC-QExactive-Orbitrap-MS.

A pooled Quality Control (QC) sample was prepared by pooling an aliquot of extract from each sample.

### 4.4. Polar Metabolome Analysis

The analysis of polar metabolome was performed on a Thermo Ultimate RS 3000 coupled online to a Q-Exactive hybrid quadrupole Orbitrap mass spectrometer (Thermo Fisher Scientific, Bremen, Germany) equipped with a heated electrospray ionization probe (HESI II). The MS was calibrated by Thermo calmix Pierce™ calibration solutions in both polarities. Separation was performed using a SeQuant^®^ ZIC^®^-HILIC column (100 × 2.1 mm; 3.5 µm, 100Å) protected with a SeQuant^®^ ZIC^®^-HILIC Guard precolumn (20 × 2.1 mm) (Supelco^®^ Sigma Aldrich). The column temperature was set at 40 °C, and the flow rate was 0.350 mL/min. The mobile phase was (A) 10 mM CH_3_COONH_4_ in H_2_O/ACN (95:5 *v*/*v*) and (B) 10 mM CH_3_COONH_4_ in H_2_O/ACN (5:95 *v*/*v*). The following gradient was employed: 0 min, 100% B, isocratic for 1.5 min; 1.51–3.5 min, 100–70% B; 3.51–9 min, 70–50% B; 9.01–9.50 min, 50–20% B, isocratic for 2 min; returning to 100% in 0.1 min. At the end of the gradient, 5 µL were injected. Full MS (80–800 *m*/*z*) and data-dependent MS/MS were performed at a resolution of 35,000 and 15,000 FWHM, respectively, and normalized collision energy (NCE) values of 10, 20 and 30 were used. Source parameters were as follows: sheath gas pressure, 50 arbitrary units; auxiliary gas flow, 13 arbitrary units; spray voltage, +3.5 kV, −2.8 kV; capillary temperature, 310 °C; and auxiliary gas heater temperature, 300 °C. Three replicates of each sample were performed in each polarity, and QC was randomly inserted in the batch to monitor system stability over time.

### 4.5. Data Analysis

Q-Exactive MS/MS data analysis was performed using MS-DIAL v4.48 (http://prime.psc.riken.jp/compms/msdial/main.html accessed on 30 October 2023). Thermo RAW. data files were converted to ABF format using the Reifycs Abf (Analysis Base File) converter. Subsequently, the alignment of Profile Q-Exactive files was performed with MS tolerance set at 0.05 for MS and MS/MS, with a retention time tolerance of 0.2 min. The minimum peak height for detection was set to 10,000 amplitude value. Metabolite identification was performed using internal libraries ESI POS and ESI NEG All_Public_MS/MS. The spectra were processed in positive mode using [M + H]^+^, [M + Na]^+^, [M + K]^+^, [M + H–H_2_O]^+^ and [M + NH_4_]^+^ ions, while [M–H]^−^, [M + Cl]^−^, [M + CH_3_COO]^−^ and [M–H_2_O]^−^ were processed in negative mode. All metabolite annotations were based on mass accuracy, isotopic pattern and spectral matching, and rev.dot product, and the score cut-off was 80%. All reported spectral matches were manually revised for correct assignment. Data were normalized based on sumTIC, and were log-transformed and autoscaled before statistical analysis.

### 4.6. Proteome Analysis

A mount of 100 µg of proteins, determined using Bradford reagent, was digested using EasyPep™ MS Sample Prep Kits (Thermo Fisher). After digestion, peptides were labeled using a TMT 2plex kit. Analysis was performed by nanoflow ultra-high performance liquid chromatography–high resolution mass spectrometry using an Ultimate 3000 nanoLC (Thermo Fisher Scientific, Bremen, Germany) coupled to an Orbitrap Lumos tribrid mass spectrometer (Thermo Fisher Scientific) with a nanoelectrospray ion source (Thermo Fisher Scientific). 1 μL of the digest was initially trapped on a PepMap trap column for 1.50 min at a flow rate of 40 μL/min (Thermo Fisher), and then peptides were loaded and separated onto a C18 reversed-phase column (150 mm × 75 μm I.D, 3.0 µm, 100 Å, EasySpray, Thermo). Mobile phases were (A) 0.1% HCOOH in water *v*/*v*, and (B) 0.1% HCOOH in ACN/Water *v*/*v* 80/20, and a linear 90 min gradient was performed. Each sample was analyzed by LC-MS/MS in duplicate. MS data were acquired using a data-dependent method, dynamically choosing the most abundant precursor ions from the survey scan (375–1500 *m*/*z*) using HCD fragmentation. Survey scans were acquired at a resolution of 120,000 at *m*/*z* 200. Unassigned precursor ion charge states as well as singly charged species were excluded. The isolation window was set to 3 Da normalized, and collision energies (NCEs) of 27 were applied. Maximum ion injection times for MS (OT) and the MS/MS (OT) scans were set to auto and 60 ms, respectively, and ACG values were set to standard. The dynamic exclusion was set to 30 s. The raw data were analyzed using Proteome Discoverer 2.5 with the Sequest HT-like search engine. The following parameters were used: enzyme trypsin, missed cleavages max 2, precursor mass tolerance 10 ppm and fragment mass tolerance 0.02 Da. Carbamidomethylcysteine was used as fixed modification, while methionine oxidation, TMT 2plex/+229.163 Da (K) and peptide N-Terminus TMT 2plex/+229.163 Da (Any N-Terminus) was used as the variable. Proteins were considered identified with at least one unique peptide setting a false discovery threshold of <1%. The TMT quantification was performed using Proteome Discoverer 2.5 software. This workflow was based on the reporter ion, measured by the mass spectrometer, which generates a different low-mass peak for each sample. As a result, the peak height for each reporter denotes the relative amount of peptide originating from each of the labeled samples. The TMT data normalization was performed by Reporter Ions Quantifier node of Proteome Discoverer 2.5 software. To equalize peptide abundances across all TMT channels and to correct for differences in sample loading, a normalization to the total peptide amount was applied. After calculating the normalization factors, the Reporter Ions Quantifier node-normalized quan spectra (for reporter ion quantification) were obtained by dividing abundances with the normalization factor over all samples.

### 4.7. Western Blotting

Protein extraction from EC and healthy endometrial tissue was performed as previously described [46]. Briefly 10 mg from EC and healthy tissue was homogenized in lysis buffer (1% NP-40, 50 mM Tris-HCl (pH 8.0), NaCl 150 mM) with Phosphatase Inhibitor Cocktail Set II 1× (Millipore, Burlington, VT, USA), 2 mM phenylmethylsulfonyl fluoride (PMSF), and 1 mM benzamidine. Protein concentration was determined by Bradford assay.

Western blotting analysis confirmed the altered abundance of NPEPPS in 10 EC and 10 healthy samples, as previously described [47]. For Western blotting analysis, 30 µg of tissue lysate was loaded on 4–20% precast gel and then transferred to a nitrocellulose membrane. After protein transfer, the membrane was blocked by treatment with 5% defatted milk in TBS-tween 20 and incubated overnight at 4 °C with 1:1000 diluted primary rabbit polyclonal antibody against NPEPPS. After primary antibody incubation, the membranes were washed 3 times with TBS-Tween 0.05% and incubated with HRP-conjugated anti-rabbit IgG (1:3000, Sigma-Aldrich; Merck KGaA, Darmstadt, Germany). Protein band signal visualization was performed by using SuperSignal West Pico Chemiluminescent (Thermo Fisher Scientific Inc., Ottawa, ON, Canada). The intensities of the immunostained bands were normalized with the total protein intensities measured by staining the membranes from the same blot with a Red Ponceau solution (Sigma-Aldrich, St. Louis, MO, USA).

### 4.8. Bioinformatic Analysis

Proteins identified by MS were analyzed by gProfiler bioinformatic tool for protein characterization according to their molecular function and protein class. The bio-functions were generated via Ingenuity Pathway Analysis (IPA) with a significance of *p* < 0.01, as previously described [48]. For the filter summary, we considered an association with high confidence, and experimentally observed. For metabolites, we used Metaboanalyst 5 for pathway enrichment and chemical main class identification. Venn Diagram tool was used for data correlation in multi-omics.

### 4.9. Statistical Analysis

Differences were considered significant between cell lines type 1 and type 2 when metabolites showed a fold change of ±1.5 and satisfied the *t*-test (*p* < 0.05), while proteins were considered significant when they showed a fold change ±1.2 and satisfied the *t*-test (*p* < 0.05). All analyses were conducted using Stata/IC 16.1.

## 5. Conclusions

Our multi-omics study has allowed us to identify different common pathways between the two types of EC. The identified pathways are clues that show the mechanisms in common between the two cancer types. In our opinion, this is quite relevant as it can open the way for new functional studies to better understand the mechanisms involved in EC, especially for the development of new therapies.

## Figures and Tables

**Figure 1 ijms-24-16057-f001:**
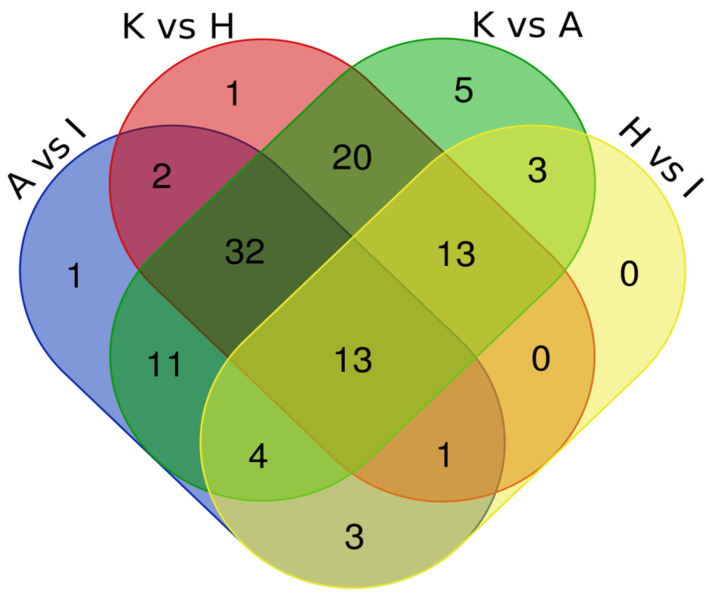
Intersection of metabolomics data by Venn Diagram.

**Figure 2 ijms-24-16057-f002:**
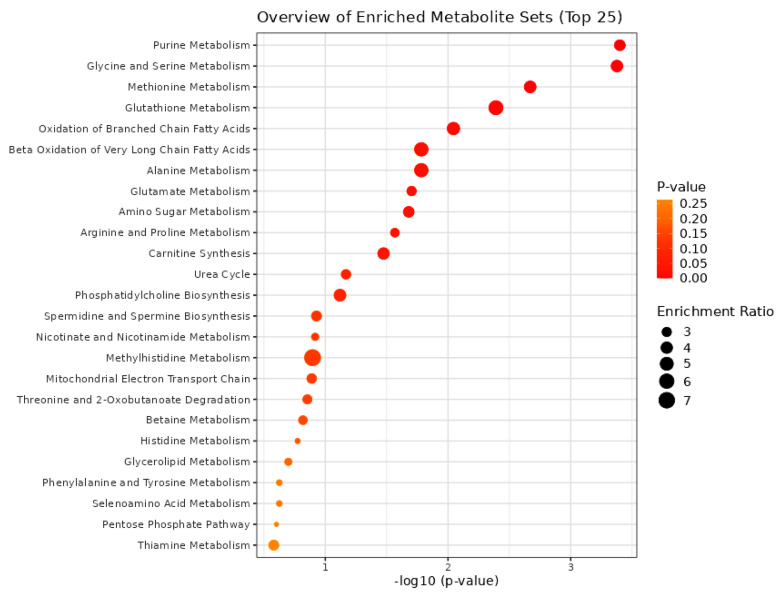
Overview of the metabolic pathways using the MetaboAnalyst bioinformatic tool. Result of the Metabolic Set Enrichment Analysis (MSEA) showing the threefold enrichment of several amino acid and fatty acid pathways.

**Figure 3 ijms-24-16057-f003:**
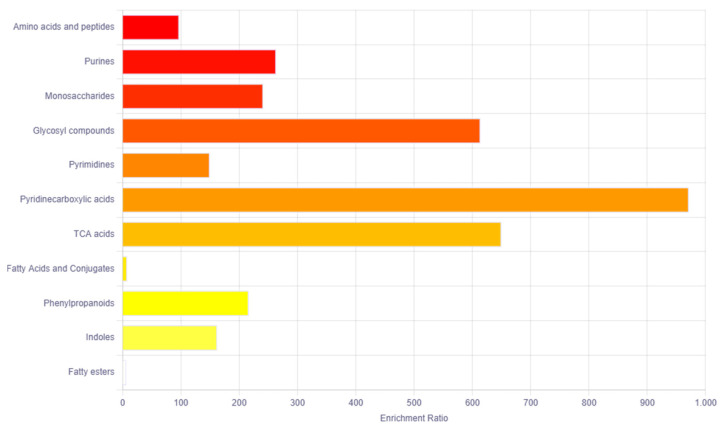
Overview of the main chemical classes using the MetaboAnalyst bioinformatic tool. Result of the Metabolic Set Enrichment Analysis (MSEA) showing the threefold enrichment of amino acids, purines, monosaccharides and glycosyl compounds.

**Figure 4 ijms-24-16057-f004:**
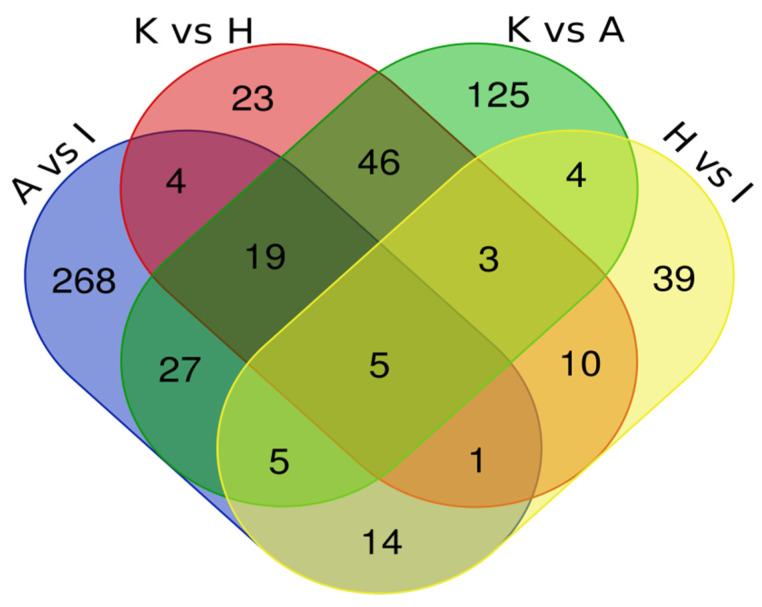
Intersection of proteomics data by Venn Diagram.

**Figure 5 ijms-24-16057-f005:**
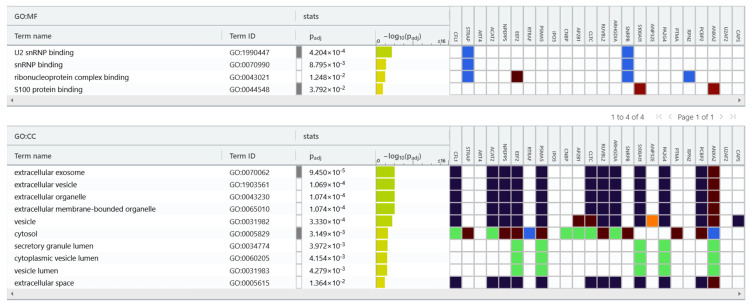
gProfiler classification of the EC proteins identified in the study. Result of proteins set enrichment analysis showing the enrichment of molecular function (MF): U2 snRNP binding, snRNP binding, ribonucleoprotein complex binding, S100 protein binding and cellular component (CC): extracellular exosome, extracellular vesicle, extracellular organelle, extracellular membrane-bounded organelle, vesicle, vesicle lumen, extracellular space, cytosol, and secretory granule lumen.

**Figure 6 ijms-24-16057-f006:**
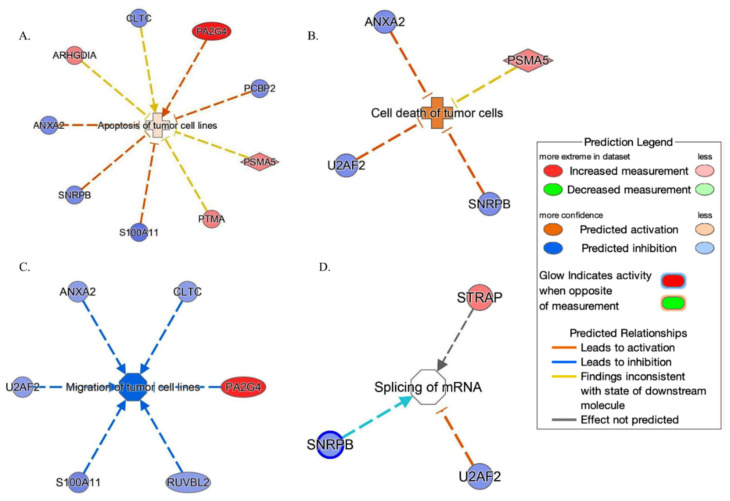
Network build-up from one of the most significant bio-functions: (**A**) the apoptosis of tumor cell lines; (**B**) the cell death of tumor cells; (**C**) the migration of tumor cell lines; (**D**) the splicing of mRNA.

**Figure 7 ijms-24-16057-f007:**
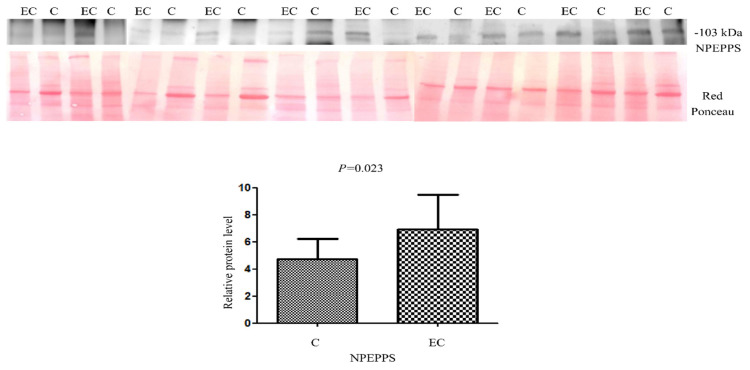
Western blot analysis was utilized to confirm the alteration of protein NPEPPS in endometrial cancer (EC) when compared to the normal endometrium (C). The intensity of immunostained bands was normalized against the total protein intensities measured from the same blot stained with Red Ponceau. Results are displayed as a histogram (*p* < 0.05), and each bar represents mean ± standard deviation.

**Table 1 ijms-24-16057-t001:** Dysregulated metabolites identified in four comparisons between type 1 and type 2 EC cell lines.

Cell Line Names	Number of Significant Metabolites	Metabolites Up/ Downregulated
ANCA (type 1 metastatic)/ISHIKAWA (type 1 non-metastatic)	67	2 up/65 down
KLE (metastatic type 2)/HEC1A (non-metastatic type 2)	82	76 up/6 down
KLE (type 2 metastatic)/ANCA (type 1 metastatic)	101	98 up/3 down
HEC1A/(type 2 not metastatic)/ISHIKAWA (type 1 non-metastatic)	37	23 up/14 down

Up-fold change ≥ 1.5. Down-fold change ≤ 0.69.

**Table 2 ijms-24-16057-t002:** Different abundance in the metabolites identified by mass spectrometry in all comparisons.

Metabolite	ANCA/ISHIKAWA	KLE/ANCA	KLE/HEC	HEC1A/ISHIKAWA
	F.C. (*p*-Value)	F.C. (*p*-Value)	F.C. (*p*-Value)	F.C. (*p*-Value)
Xanthine	↓0.015 (0.0054)	↑15.43 (0.0002)	↓0.566 (0.029)	↓0.424 (0.037)
N-Acetylaspartic acid	↓0.089 (0.002)	↑25.566 (7.29 × 10^−8^)	↑8.726 (2.52 × 10^−7^)	↓0.261 (0.004)
Methyl-2-hydroxyisobutyric acid	↓0.395 (0.0007)	↑35.412 (0.0002)	↑23.842 (0.0002)	↓0.587 (0.011)
FA 20:4	↓0.568 (0.040)	↑2.179 (0.016)	↑4.049 (0.005)	↓0.305 (0.010)
Butyryl carnitine	↓0.249 (0.020)	↑313.21 (2.27 × 10^−5^)	↑13.505 (3.87 × 10^−5^)	↓5.778 (0.019)
Acetylcarnitine	↓0.479 (0.010)	↑223.24 (1.86 × 10^−5^)	↑8.373 (4.47 × 10^−5^)	↑12.796 (0.005)
4-Aminobutyric acid	↓0.147 (0.010)	↑67.269 (1.05 × 10^−5^)	↑4.152 (0.0001)	↑2.387 (0.026)
Cystathionine	↓0.243 (0.018)	↑56.262 (0.0002)	↑27.491 (0.0002)	↓0.498 (0.088)
Glucosamine-6-phosphate	↓0.098 (0.001)	↑46.353 (2.92 × 10^−5^)	↑17.422 (3.97 × 10^−5^)	↓0.261 (0.005)
S-Adenosylhomocysteine	↓0.046 (0.002)	↑7.161 (0.001)	↑1.813 (0.011)	↓0.182 (0.004)
Threonine	↓0.318 (0.004)	↑4.604 (0.0004)	↑2.124 (0.002)	↓0.689 (0.049)
Methionine sulfoxide	↓0.468 (0.030)	↓0.291 (0.041)	↓0.064 (0.002)	↑2.105 (0.022)
Fumaric acid	↓0.151 (0.002)	↑18.938 (0.001)	↑7.799 (0.001)	↓0.368 (0.006)
Indoleacetic acid	↓0.631 (0.019)	↑3.65 (0.002)	↑2.054 (0.008)	N.P.
Pyroglutamic acid	↓0.151 (0.002)	↑36.831 (1.35 × 10^−5^)	↑4.940 (0.001)	N.P.
Guanosine	↓0.568 (0.040)	↑22.307 (2.36 × 10^−6^)	↑20.994 (4.99 × 10^−6^)	N.P.
Nicotinamide	↓0.225 (0.022)	↑25.41 (1.9 × 10^−5^)	↑4.376 (4.71 × 10^−5^)	N.P.
Uric acid	↓0.283 (0.007)	↑10.992 (3.47 × 10^−5^)	↑4.906 (8.05 × 10^−5^)	N.P.
Vaccenic acid	↓0.442 (0.012)	↑4.891 (0.0004)	↑3.773(0.001)	N.P.
Glyceric acid	↓0.184 (0.030)	↑3.405 (0.002)	↑1.967 (0.012)	N.P.
Proline	↓0.262 (0.001)	↑13.604 (2.82 × 10^−7^)	↑3.04 (1.76 × 10^−5^)	N.P.
Methionine	↓0.167 (0.030)	↑25.05 (0.0002)	↑5.857(0.002)	N.P.
Norleucine	↓0.212 (0.040)	↑14.044 (0.001)	↑2.588 (0.036)	N.P.
Inosine	↓0.206 (0.016)	↑33.445 (1.15 × 10^−6^)	↑10.470 (4.04 × 10^−6^)	N.P.
Glucose phosphate	↓0.092 (0.010)	↑84.07 (2.86 × 10^−9^)	↑14.375 (8.21 × 10^−7^)	N.P.
Carnosine	↓0.336 (0.008)	↑15.182 (1.99 × 10^−5^)	↑6.516 (4.11 × 10^−5^)	N.P.
D-Ribose 5-phosphate	↓0.146 (0.004)	↑15.814 (0.0002)	↑4.373 (0.0015)	N.P.
Alanine	↓0.248 (0.0003)	↑6.261 (1.62 × 10^−5^)	↑2.001(0.0002)	N.P.
N-Acetylaspartylglutamic acid	↓0.148 (0.037)	↑67.34 (0.002)	↑13.188 (0.003)	N.P.
Uridine	↓0.228 (0.016)	↑22.381 (0.003)	↑3.661 (0.003)	N.P.
Hydroxyphenyllactic acid	↑2.073 (0.020)	↑11.11 (0.0002)	↑22.808(0.0002)	N.P.
Hypoxanthine	↓0.139 (0.011)	↑5.459 (0.0001)	↑6.753 (0.0002)	N.P.
N-Acetyl-methionine	↓0.09 (0.006)	↑78.474 (0.0003)	↑9.185 (0.001)	N.P.
Adenosine diphosphate	↓0.103 (0.030)	↑178.687 (0.0001)	↑20.871 (0.0002)	N.P.
Carnitine	↓0.162 (0.030)	↑133.546 (0.0001)	↑12.241 (7.74 × 10^−5^)	N.P.
Cytidine	↓0.167 (0.015)	↑10.424 (0.006)	↑3.583 (0.017)	N.P.
Cytosine	↓0.092 (0.030)	↑18.731 (0.027)	↑6.08 (0.041)	N.P.
N-Acetylneuraminic acid	↓0.282 (0.030	↑76.425 (0.0002)	↑11.615 (0.0003)	N.P.
Propionylcarnitine	↓0.17 (0.020)	↑193.523 (0.003)	↑23.871 (0.003)	N.P.
Fructose	↓0.067 (0.007)	↑64.354 (0.001)	↑8.418 (0.002)	N.P.
Phenylalanine	↓0.19 (0.036)	↑13.149 (0.001)	↑3.138 (0.014)	N.P.
N6-Acetyllysine	↓0.3 (0.002)	↑4.9365 (0.002)	↑2.027 (0.017)	N.P.
N-Acetylysine	↓0.587 (0.021)	↑4.792 (0.0004)	↑3.335 (0.001)	N.P.
5′-Methylthioadenosine	↓0.367 (0.010)	↑36.743 (3.68 × 10^−5^)	↑6.792 (0.0001)	N.P.

N.P.: Not present. F.C.: fold change. ↑: up-regulated. ↓: down-regulated.

**Table 3 ijms-24-16057-t003:** Dysregulated proteins identified in four comparisons between four different type 1 and type 2 EC cell lines.

Cell Name	Number of Significant Proteins	Proteins Up/ Downregulated
ANCA (type 1 metastatic)/ISHIKAWA (type 1 non-metastatic)	340	309 up/31 down
KLE (metastatic type 2)/HEC1A (non-metastatic type 2)	111	55 up/56 down
KLE (type 2 metastatic)/ANCA (type 1 metastatic)	234	122 up/112 down
HEC1A (type 2 not metastatic)/ISHIKAWA (type 1 non-metastatic)	81	40 up/41 down

Up-fold change ≥ 1.2. Down-fold change ≤ 0.85.

**Table 4 ijms-24-16057-t004:** Different abundance proteins identified by mass spectrometry present in all comparisons.

Genes Accession	ANCA/ISHIKAWA	KLE/ANCA	KLE/HEC	HEC1A/ISHIKAWA
	F.C. (*p*-Value)	F.C. (*p*-Value)	F.C. (*p*-Value)	F.C. (*p*-Value)
CFL1 P23528	↓0.761 (0.005)	↓0.676 (0.001)	↓0.794 (0.019)	↓0.851 (0.049)
STRAP Q9Y3F4	↑2.074 (7.91 × 10^−5^)	↑1.656 (0.038)	↓1.963 (0.020)	↓0.843 (0.044)
ART3 Q13508	↑1.217 (0.031)	↑1.474 (0.002)	↑1.213 (0.024)	↑1.215 (0.048)
ACAT2 Q9BWD1	↑2.143 (5.73 × 10^−7^)	↓0.205 (0.001)	↓0.112 (0.039)	↑1.82 (0.043)
NPEPPS P55786	↑2.561 (5.25 × 10^−5^)	↑1.62 (0.002)	↑1.369 (0.0015)	↑1.183 (0.032)
EEF2 P13639	↓0.179 (2.87 × 10^−5^)	↓0.765 (0.046)	↓0.738 (0.026)	N.P.
RTRAF Q9Y224	↑2.200 (0.0003)	↓0.328 (0.006)	↓0.354 (0.029)	N.P.
PSMA5 P28066	↑1.696 (0.022)	↓0.744 (0.044)	↓0.672 (0.047)	N.P.
IPO5 O00410	↑1.915 (0.026)	↓0.343 (0.004)	↓0.369 (0.005)	N.P.
CNBP P62633	↑2.125 (0.003)	↑1.847 (0.045)	↑1.901 (0.035)	N.P.
AP2B1 P63010	↑2.198 (0.0002)	↓0.452 (0.005)	↓0.422 (0.008)	N.P.
CLTC Q00610	↑1.294 (0.003)	↓0.698 (0.001)	↓0.717 (0.002)	N.P.
RUVBL2 Q9Y230	↑2.067 (6.17 × 10^−6^)	↓0.559 (0.005)	↓0.501 (0.019)	N.P.
ARHGDIA P52565	↑1.769 (0.017)	↑1.302 (0.002)	↑1.47 (0.023)	N.P.
SNRPB P14678	↓0.459 (0.011)	↓0.671 (0.012)	↓0.77 (0.030)	N.P.
S100A11 P31949	↑1.785 (0.011)	↓0.265 (0.005)	↓0.249 (0.005)	N.P.
ANP32E Q9BTT0	↑2.315 (8.1 × 10^−8^)	↓0.153 (1.6 × 10^−5^)	↓0.151 (0.002)	N.P.
PA2G4 Q9UQ80	↑1.271 (0.038)	↑1.238 (0.037)	↑1.323 (0.022)	N.P.
PTMA P06454	↓0.855 (0.005)	↑1.545 (0.001)	↑1.459 (0.002)	N.P.
RPN2 P04844	↑1.657 (0.035)	↓0.331 (0.013)	↓0.38 (0.028)	N.P.
PCBP2 Q15366	↑1.930 (0.0004)	↓0.462 (0.005)	↓0.537 (0.001)	N.P.
ANXA2 P07355	↑1.294 (0.016)	↓0.613 (0.001)	↓0.644 (0.007)	N.P.
U2AF2 P26368	↑2 (7.06 × 10^−5^)	↓0.222 (0.001)	↓0.231 (0.039)	N.P.
CAPS Q9ULU8	↑2.033 (1.71 × 10^−8^)	↓0.428 (0.020)	↓0.293 (0.038)	N.P.

F.C.: Fold change. N.P.: not present. ↑: up-regulated. ↓: down-regulated.

**Table 5 ijms-24-16057-t005:** Pathways identified by multi-omics analysis in both EC cell types 1 and 2.

Pathway Name	*p*-Value
ABC transporters	4.87 × 10^−8^
Alanine, aspartate and glutamate metabolism	2.69 × 10^−7^
Mineral absorption	1.42 × 10^−6^
Purine metabolism	3.40 × 10^−6^
Central carbon metabolism in cancer	4.53 × 10^−6^
Cysteine and methionine metabolism	6.24 × 10^−6^
Aminoacyl-tRNA biosynthesis	8.43 × 10^−6^
Protein digestion and absorption	2.43 × 10^−5^
Synaptic vesicle cycle	5.08 × 10^−4^
Glycine, serine and threonine metabolism	5.08 × 10^−4^
Butanoate metabolism	0.0030014
Valine, leucine and isoleucine biosynthesis	0.0055171
Nicotinate and Nicotinamide metabolism	0.0064624
Glyoxylate and dicarboxylate metabolism	0.0064624
Lysine degradation	0.011893
Pyrimidine metabolism	0.013939
Endocrine and other factor-regulated calcium reabsorption	0.026315
Phosphonate and phosphinate metabolism	0.027938
Beta-alanine metabolism	0.027938
Pentose phosphate pathway	0.0296
Pyruvate metabolism	0.033923
Phenylalanine metabolism	0.040361

## Data Availability

The data presented in this study are available upon request from the corresponding author. The data are not publicly available due to ethical reasons.

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
