# Peer review of "A Multi-Omics Approach Revealed Common Dysregulated Pathways in Type One and Type Two Endometrial Cancers"

_ijms, 2023, doi:10.3390/ijms242216057_

Round 1

Reviewer 1 Report

Comments and Suggestions for Authors

Page1, line 45: Word 'positive' is ambiguous and it must be substitute to 'dependent' in comparison with in-dependent which shown in next sentence.

Page 1, line 46: Word 'high-grade' should be explained in comparison with low-grade, and intermediate-grade (or combination). Grade in endometrial cancer meant WHO grading in pathological architecture components. However, in this sentence, it seemed to be poor prognostic cancer. In ovarian cancer, high-grade serous meant serous type carcinoma with good sensitivity to platinum agents. The confusion would be occurred between poor prognosis and well controllable cancer with its good platinum sensitivity. That is, the worst prognostic tumors such as carcinosarcomas, and serous type of endometrium would not be called as high-grade.

Author Response

Page1, line 45: Word 'positive' is ambiguous and it must be substitute to 'dependent' in comparison with in-dependent which shown in next sentence.

Our reply: We fixed the sentence.

Page 1, line 46: Word 'high-grade' should be explained in comparison with low-grade, and intermediate-grade (or combination). Grade in endometrial cancer meant WHO grading in pathological architecture components. However, in this sentence, it seemed to be poor prognostic cancer. In ovarian cancer, high-grade serous meant serous type carcinoma with good sensitivity to platinum agents. The confusion would be occurred between poor prognosis and well controllable cancer with its good platinum sensitivity. That is, the worst prognostic tumors such as carcinosarcomas, and serous type of endometrium would not be called as high-grade.

Our reply: We deleted the phrase to avoid confusion.

Reviewer 2 Report

Comments and Suggestions for Authors

Dear author,

the article is very clear and addresses and the multi-omits strategy was well conducted. My minor suggestions are:

Remove the two dots on line 25 of the abstract.

Improve the introduction to give a bit more background on the subject

Figure 2, Figure 3 and Figure 8 lack legends, please add a short legend to explain what is shown in the figure. 

The discussion needs an improvement. I believe the explanations prided are a bit shallow. Please improve also mentioning the drawbacks of the current work and what could be done to improve it.

Author Response

Reviwer 2

Remove the two dots on line 25 of the abstract.

Our reply: We fixed the dots.

Improve the introduction to give a bit more background on the subject

Our reply: We improved the introduction.

Figure 2, Figure 3 and Figure 8 lack legends, please add a short legend to explain what is shown in the figure.

Our reply: We added a short legend to Figure 2, Figure 3 and Figure 8.

The discussion needs an improvement. I believe the explanations prided are a bit shallow. Please improve also mentioning the drawbacks of the current work and what could be done to improve it.

Our reply: We improved the discussion as recommended by the Reviewer.

Reviewer 3 Report

Comments and Suggestions for Authors

Utilizing multi-omics data to understand complex diseases such as cancer is of great importance. In this manuscript, the authors tried to analyse multi-omics data of endometrial cancer cells lines to find important pathways related to different types of endometrial cancers. 

My major comments:

1. It seems the fold-change cut-offs for metabolomics and proteomics are quite arbitrary.  Could the authors explain why those different cut-offs were used or selected?

2. When the overlapped metabolites were few, the authors chose to use three comparisons? Why was the last comparison (HEC vs ISHIKAWA) excluded? 

3. In the western blotting results, only one protein with p value less than 0.05. I don't think the authors can claim that they validated three proteins.

Author Response

Reviewer 3

It seems the fold-change cut-offs for metabolomics and proteomics are quite arbitrary.  Could the authors explain why those different cut-offs were used or selected?

Our reply: It should always be taken into account that some of the significantly changed proteins and metabolites reported are due to natural variation. Hence, to reduce the probability of these false positives, the selection of variable proteins and metabolites should be performed by imposing a requirement of a strict fold change difference in addition to the use of statistical criteria (Renato Millioni, Lucia Puricelli, Stefano Sbrignadello, Elisabetta Iori, Ellen Murphy, Paolo Tessari. Operator- and software-related post-experimental variability and source of error in 2-DE analysis. Amino Acids (2012) 42:1583–1590).

Increasing data in the literature report the use of a ±1.5-fold increase or decrease in metabolomics as acceptable to reduce the probability of false positives. (Huaxu Yu Shipei Xing  Lorenz Nierves Philipp F Lange Tao Huan. Fold-Change Compression: An Unexplored But Correctable Quantitative Bias Caused by Nonlinear Electrospray Ionization Responses in Untargeted Metabolomics. Anal Chem. 2020 May 19;92(10):7011-7019), (Kristin D. Gerson, Jingqiu Liao, Clare McCarthy, Heather H. Burris, Tal Korem, Maayan Levy, Jacques Ravel,  Michal A. Elovitz. A non-optimal cervicovaginal microbiota in pregnancy is associated with a distinct metabolomic signature among non-Hispanic Black individuals. Sci Rep. 2021 Nov 23;11(1):22794),  (Samit Ganguly, David Finkelstein, Timothy I Shaw, Ryan D Michalek, Kimberly M Zorn, Sean Ekins, Kazuto Yasuda, Yu Fukuda, John D Schuetz, Kamalika Mukherjee, Erin G Schuetz. Metabolomic and transcriptomic analysis reveals endogenous substrates and metabolic adaptation in rats lacking Abcg2 and Abcb1a transporters. PLoS One. 2021;16:e0253852).

Moreover, in TMT analysis, we used a ±1.2-fold in protein expression to reduce the probability of false positives. Literature data supports our fold change choice (Dong-Fang Li, Zhao-Hui Cui, Lu-Yang Wang, Kai-Hui Zhang, Le-Tian Cao, Shuang-Jian Zheng, Long-Xian Zhang. Tandem mass tag (TMT)-based proteomic analysis of Cryptosporidium andersoni oocysts before and after excystation. Parasit Vectors. 2021 Dec 18;14(1):608.), (Oliver Serang, A. Ertugrul Cansizoglu, Lukas Käll, Hanno Steen, Judith A. Steen. Nonparametric Bayesian evaluation of differential protein quantification. J Proteome Res. 2013 Oct 4; 12(10): 10.1021/pr400678m), (Aidan Huang, Meiyu Zhang, Taijie Li, Xue Qin. Serum Proteomic Analysis by Tandem Mass Tags (TMT) Based Quantitative Proteomics in Gastric Cancer Patients. Clin Lab. 2018 May 1;64(5):855-866).

  1. When the overlapped metabolites were few, the authors chose to use three comparisons? Why was the last comparison (HEC vs ISHIKAWA) excluded?

Our reply: We thank the Reviewer for the question. In the second overlap metabolites analysis we excluded the fourth pairing (HEC vs ISHIKAWA) because using the 3 pairings (ANCA vs ISHIKAWA, KLE vs ANCA, KLE vs HEC) the number of metabolites identified was higher (32 metabolites) than in the first analysis with 4 pairings (13 metabolites).

  1. In the western blotting results, only one protein with p value less than 0.05. I don't think the authors can claim that they validated three proteins.

Our reply: As asked by the reviewer we have added sentences to emphasize that the proteins STRAP and ART3 are not significant.

Round 2

Reviewer 3 Report

Comments and Suggestions for Authors

Since one protein was validated by western blotting with significant p values, I suggest the authors to rephase "Western blotting analysis on 10 patients with type 1 and type 2 EC and 10 endometria samples confirmed the altered abundance of STRAP, ART3 and NPEPPS." in the abstract. And also rephase “At last, from the five (CFL1, STRAP, ART3, ACAT2, NPEPPS) common proteins identified by bioinformatic tools, we validated three of them (NPEPPS, ART3, STRAP).” in the discussion on page 10. The authors should remove STRAP and ART3 as the p values for these two proteins are far away from 0.05.

Why are 10 EC vs 10 healthy controls enough to validate NPEPPS but not enough for STRAP and ART3?

Author Response

Since one protein was validated by western blotting with significant p values, I suggest the authors to rephase "Western blotting analysis on 10 patients with type 1 and type 2 EC and 10 endometria samples confirmed the altered abundance of STRAP, ART3 and NPEPPS." in the abstract. And also rephase “At last, from the five (CFL1, STRAP, ART3, ACAT2, NPEPPS) common proteins identified by bioinformatic tools, we validated three of them (NPEPPS, ART3, STRAP).” in the discussion on page 10. The authors should remove STRAP and ART3 as the p values for these two proteins are far away from 0.05.

Why are 10 EC vs 10 healthy controls enough to validate NPEPPS but not enough for STRAP and ART3?

Our reply: We fixed the manuscript as requested by the Reviewer. As requested, we removed the mention to STRAP and ART3 from the manuscript. The number of samples was enough to validate the NPEPPS, because the abundance of the protein in all samples goes in the same direction and the Mann-Whitney test was significant. While for the other two proteins, the abundance does not go in the same direction and the Mann-Whitney was not statistically significant.